# CAE Prediction for Compression Behavior in Multi-Stacked Packages with an EPS-Based Cushioning System: Modeling of Compression and Compressive Creep Behavior

**Jong Min Park [1], Gun Yeop Lee [2], Dong Hyun Kim [1] and Hyun Mo Jung [3,\*]**

[1]  Department of Bio-Industrial Machinery Engineering, Pusan National University, Miryang 50463, Republic of Korea
[2]  Appliance Advanced Technology R&D Group, LG Electronics, Changwon 51533, Republic of Korea
[3]  Department of Logistic Packaging, Kyungbuk Science College, Chilgok 39913, Republic of Korea
\*  Correspondence: hmjung@kbsc.ac.kr

**Abstract:** The compression and compressive creep behavior of target shipping containers, which are material properties based on finite element analysis, and the lifetime and load-sharing rate, were analyzed in this study to develop a computer-aided engineering prediction technology for predicting the multi-stage compression behavior of three target packages with different logistics conditions. In the experiment performed in the study, the relative humidity levels were 50%, 70%, and 90%, with creep measurements performed for 12 h for a combination of three levels of applied load and relative humidity. Using the nonlinear model of the stress–strain and creep behavior of the target shipping container, the lifetime was analyzed by estimating the average creep rate of the target shipping container. The load-sharing rate for each logistics situation of the target packages was also analyzed. The reduction rate of the compression strength of the container with respect to the increase in relative humidity was greater in the 'horizontal long' container than in the 'vertical long' container. As the applied load increased, the rate of increase in the average creep rate increased, i.e., the higher the applied load, the larger the difference in the average creep rate with respect to the relative humidity. The lifetime estimated from the failure strain and average creep rate of the container gradually decreased as the applied load increased at all relative humidity levels. However, as the applied load increased, the difference with respect to the relative humidity tended to decrease. In the target packages used in this study, the ratio of the load-sharing rate between the shipping container and an expanded polystyrene cushioning material was determined to be 2%:98%, with most of the stacking load applied to the product through the cushioning material.

**Keywords:** corrugated fiberboard container; creep behavior; EPS-based cushioning system; lifetime; secondary creep rate; load-sharing rate

## 1. Introduction

Heavy home appliances (e.g., refrigerators, washing machines, and air conditioners) are usually cushioned by an expanded polystyrene (EPS)-based cushioning system and packaged in a shipping container (mostly corrugated fiberboard containers). These packages are unitized on a pallet in a certain quantity (palletized-unit load) and are subject to logistics operations, such as transportation, storage, and handling. They are commonly stacked in multiple stages during storage. In such a situation, where the package is subject to long-term stacking loads, the shipping container and EPS-based cushioning system in a package undergo creep deformation. The longer the package is stacked on the lower layer, the more severe the creep deformation.

In a warehouse, the cumulative creep deformation of multi-stacked packages may lead to an inclination of the unitized load and even collapse. This phenomenon occurs more conspicuously by the action of the rotational moment owing to an eccentric load

when the center of gravity is eccentric to one side in the package of heavy goods, such as refrigerators and washing machines.

In general, the creep behavior of a viscoelastic material has been described to have three phases, as shown in Figure 1: (1) primary (transient) creep, (2) secondary (steady-state) creep, and (3) tertiary (accelerating) creep. In the primary phase, the strain increases with a decrease in the strain rate. In the second phase, the strain increases linearly with time, yielding a constant strain rate. The tertiary phase is characterized by a rapid increase in strain, rapidly leading to failure. However, the actual creep behavior of viscoelastic materials is strongly influenced by the properties of the material itself and the level of load applied, stabilizing after transient creep and leading to destruction after rapid progression through the three creep phases, or fully proceeding with the three creep phases.

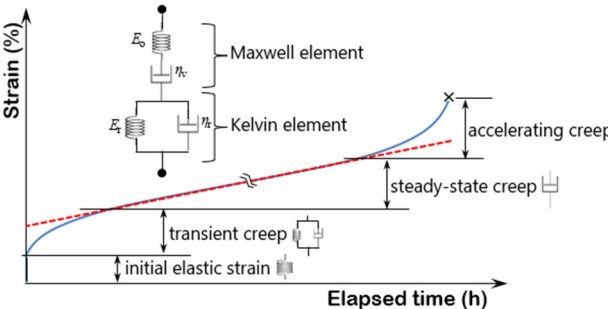

**Figure 1.** Typical creep behavior of a viscoelastic material and Burgers model.

The creep behavior, from instantaneous elastic strain to steady-state creep after loading, can be described by the Burgers model: $\varepsilon(t) = \sigma_o/E_o + \sigma_o/E_r\{1 - exp(-t/\tau_r)\} + (\sigma_o/\eta_v)t$, where $E_o$ = instantaneous elastic modulus (Pa), $E_r$ = retarded elastic modulus (Pa), $\eta_v$ and $\eta_r$ are the viscosity coefficients of the dashpot (Pa·s), and $\tau_r$ is the retardation time (=$\eta_r/E_r$)(s). The constants of this model allow for a quantitative understanding of the characteristics of the different creep phases. In the Burgers model, the Maxwell and Kelvin elements are connected in series. The spring of the Maxwell element represents the instantaneous elastic strain, whereas the dashpot represents a non-reversible viscous flow. The Kelvin element describes the delayed elasticity [1,2].

Corrugated fiberboard containers are mostly used as shipping containers for home appliances. In the case of a corrugated fiberboard container, the structure has a unique characteristic that renders its creep behavior as different from that of a simple viscoelastic material. The primary creep region of the corrugated fiberboard container is the deformation accommodated in the flattening process of the wing in the vertical direction to the applied load. The creep rate and quantity are large because the wing rotates and bends around the crease [3]. Part of the creep deformation generated in this phase is recoverable, occurring for a short time and not contributing to the failure of the container. The secondary creep region is dominated by the bulging of the container side and the load transfer to the container corner. The secondary creep rate is nearly constant; however, most of the deformation is inelastic and contributes to failure [4]. Local buckling at the corner of the container forms a hinge at the corner, resulting in overall buckling, eventually proceeding to tertiary creep and leading to the ultimate collapse of the container.

The container lifetime (time to failure) in the multi-stacked condition is significantly affected by the level of applied load (stacking load) and by the relative humidity (rh) and moisture content of the container itself (owing to the hygroscopic properties of cellulosic materials). Many researchers [5–7] have reported that cyclic humidity conditions at the same levels of the applied load lower the container lifetime compared to constant humidity conditions. However, Hussain et al. [3] reported that the effects of cyclic and constant humidity conditions differed according to the applied load level. In other words, constant humidity conditions under a high applied load level deteriorated the lifetime more than cyclic humidity conditions, with one reason for this being that the cyclic humidity condi-

tions did not reach the equilibrium moisture condition. However, they stated that testing under cyclic humidity conditions was more appropriate for the packaging industry, as most of the containers were stacked under lower applied load conditions in the supply chain of the container.

Many researchers [3,5,8] have studied the relationship between the lifetime and secondary creep rate of containers, because the majority of the lifetimes of multi-stacked containers are affected by secondary creep regions. Hussain et al. [3] calculated the slope of the linear regression equation for the data corresponding to 10–90% and 20–80% of the period through creep measurement for 21 days; the values obtained were expressed as the secondary creep rates of the failing and non-failing containers within the period, respectively. They reported that the container lifetime was inversely proportional to the secondary creep rate. Long-term stacking tests ranging from several days to several months are required to determine the secondary creep rate of corrugated fiberboard containers. Burgers et al. [9] presented a container lifetime estimation method by obtaining the average creep rate through a creep experiment of only 12 h, and correcting it to failure according to the applied load level. The method they proposed starts on the premise that creep failure of a container occurs when the creep strain reaches the quasi-static compression failure strain of the container: $T = D/R$, where $T$ = lifetime (time to fail) (min), $D$ = ASTM D642 failure deformation (mm), $R$ = average creep rate $[= R_{12} \times P/(100 - P)]$ over the entire time up to failure at the applied load level $P$, $R_{12}$ = average creep rate in the 12 h test (slope of straight line fitted to creep data), and $P$ = load expressed as a percentage of the ASTM D642 compression strength (*CS*) [10]. As a more recent study, Gray-Stuart et al. [11] analyzed how creep performance (secondary creep rate) and lifetime of container depend on the container conditions including fill and the area of the container that is exposed for moisture transfer. Regarding the secondary creep rate and lifetime of boxes containing product were significantly smaller than those of other conditions, they said that it was due to the occurrence of out-of-plane displacement due to the internal pressure imparted on the panels by the product. Holmvall [12] introduced the concept of reliability in predicting the lifetime of the container, and he said that the lifetime is indeed dependent not only on a stacking factor, but also on durability and the variations associated with the material or container.

Computer-aided engineering (CAE) prediction technology of compression behavior in multi-stacked home appliance packages equipped with an EPS-based cushioning system is an important technical evaluation tool that can prevent accidents caused by overturning of multi-stacked packages in warehouses. In addition, this tool can help reduce the development period owing to the preliminary review of multi-stacking, improvement of packaging quality, and optimization of packaging materials.

This study aimed to measure and model the compression and compressive creep behavior as material properties based on finite element analysis (FEA) of target shipping containers (corrugated fiberboard container) used for heavy home appliances, and to analyze the lifetime and load-sharing rate. As this study was conducted as part of developing a CAE prediction technology of compression behavior for unitized loads of packaged appliances with an EPS-based cushioning system, the variables and scope of the study reflected this scenario.

## 2. Experimental Design

### 2.1. Experimental Materials

The three cases shown in Table 1 were the target packages used for developing the CAE prediction technology that can predict compression behavior of the multi-stacked home appliance packages. The shipping container used in each of these three cases had a different shape, and the density of the EPS applied to the EPS-based cushioning system in each case was also different. Therefore, modeling the compression and compressive creep behavior of the packaging material for each case was essential for developing a CAE prediction technology for each of these cases.

**Table 1.** Details of the target packages.

| Division | Case 1 | | Case 2 | | Case 3 | |
|---|---|---|---|---|---|---|
| Outer size | - | L968 × W994 × H1870 mm | - | L690 × W664 × H890 mm | - | L892 × W381 × H249 mm |
| Total weight | - | 1.884 kN | - | 0.559 kN | - | 0.108 kN |
| Components | -<br>- | Outer: Container A<br>Inner: EPS cushion_(bottom) 25 kg/m$^3$, (top) 20 kg/m$^3$ | -<br>- | Outer: Container B<br>Inner: EPS cushion_(bottom) 22 kg/m$^3$, (top) 25 kg/m$^3$ | -<br>- | Outer: Container C<br>Inner: EPS cushion (right & left) 16 kg/m$^3$ |
| Stacking | - | 4 column stacking | - | 5 column stacking | - | 16 column stacking |
| Cushioning | -<br>- | Cushion area: 0.1323 m$^2$<br>Cushion thickness: 155 mm | -<br>- | Cushion area: 0.1488 m$^2$<br>Cushion thickness: 47 mm | -<br>- | Cushion area: 0.0384 m$^2$<br>Cushion thickness: 47 mm |
| Cushioning method |  | |  | |  | |

In Table 1, case 1 and 2 are two-piece upper and lower edge pad cushioning, and case 2 is two-piece side pad cushioning. In addition, the cushion area cushion thickness represents the contact area and the thickness of the cushioning material in contact with the base of the product (red dotted circle in each figure).

The detailed specifications of the shipping containers used in the target package are listed in Table 2. For Containers A and B, their sizes were too large to be tested; thus, the model test was reduced to a geometric similarity of 1/2. In the model testing, it was found the density (including mass, dimension, and area moment of inertia) of the corrugated fiberboard constituting the container between the prototype and model should also be reduced [13]. However, as it is virtually impossible to change the corrugated fiberboard specifications arbitrarily in terms of production, only the scale effect was considered in this study. Furthermore, because Containers A and B are both 'vertical long' with large heights, it was necessary to analyze the mechanical similarity between the prototype and model for the buckling stress condition. As shown in Table 3, the difference in the slenderness ratio between the prototype and model was negligible; therefore, it was considered that there was no difference in the buckling stress against axial compression between the two. Therefore, the mechanical behavior of the prototype could be determined through a model experiment.

**Table 2.** Specifications of the shipping containers used for the target packages [14].

| Division | Container A | | Container B | | Container C | |
|---|---|---|---|---|---|---|
| Outer size | - <br> - | (prototype) L968× W994 × H1870 mm <br> (model) L484× W497 × H935 mm[3] | - <br> - | (prototype) L690× W664 × H890 mm <br> (model) L345× W332 × H445 mm[3] | - | (prototype) L892× W381 × H249 mm |
| Container type | - <br> - | RSC, single-winged <br> (extremely) vertical long (H/L = 1.88; L/W = 1.03) | - <br> - | RSC, single-winged <br> (slightly) vertical long (H/L = 1.29; L/W = 1.04) | - <br> - | RSC, double-winged <br> (extremely) horizontal long (H/L = 0.28; L/W = 2.34) |
| Paper combination[1,2] | - <br> - | BB/F-DW <br> KLB175/S120/CK180/S120/K180 | - <br> - | A/F-SW <br> KLB225/CK180/KLB225 | - <br> - | A/F-SW <br> KLB175/CK180/K180 |

Note: (1) KLB225: (BS, bursting strength) 726 kPa, (RC, ring crush) 332 N, (t) 0.30 mm; KLB175: (BS) 569 kPa, (RC) 258 N, (t) 0.27 mm; K180: (BS) 353 kPa, (RC) 194 N, (t) 0.28 mm; CK180: (BS) 392 kPa, (RC) 270 N, (t) 0.27 mm; S120: (BS) 125 kPa, (RC) 82 N, (t) 0,17 mm. (2) KLB225 and KLB175: 40% UKP + 30% AOCC + 30% KOCC, KOCC = Korean old corrugated container, UKP = unbleached kraft pulp, AOCC = American old corrugated container; K180 and S120: 100% KOCC; CK180: 20% AOCC + 80% KOCC. (3) BB/F-DW (BB flute-double wall), A/F-SW (A flute-single wall), RSC (regular slotted container). (4) Geometric similarity, $\lambda = W_m/W_p = D_m/D_p = H_m/H_p = 1/2$ (*m* = model, *p* = prototype).

**Table 3.** Results of mechanical similarity analysis for buckling stress conditions between the prototype and model [15].

| Division | | $A$ (mm$^2$) | $I_{\overline{zz}}$ (mm$^4$) | $k$ (mm) | $L_e$ (mm) | $r^{(1)}$ |
|---|---|---|---|---|---|---|
| Container A | Prototype | 23,477.52 | $3.65 \times 10^9$ | 394.29 | 1870 | 4.74 |
| | Model | 11,666.28 | $4.47 \times 10^8$ | 195.80 | 935 | 4.77 |
| Container B | Prototype | 13,973.44 | $1.02 \times 10^9$ | 270.26 | 890 | 3.29 |
| | Model | 6932.64 | $1.25 \times 10^8$ | 134.08 | 445 | 3.32 |

Note: (1) $r = L_e/k$, $k = \sqrt{I_{\overline{zz}}/A}$ (*r* = slenderness ratio, $L_e$ = effective buckling length (column with pinned ends, $L_e = H$) (mm), *k* = minimum radius of gyration (mm), $I_{\overline{zz}}$ = area moment of inertia (mm$^4$) for the neutral axis. *A* = cross-sectional area (mm$^2$)).

## 2.2. Experimental Apparatus and Methods

In the uniaxial compression test for the shipping container, the loading rate of the universal testing machine was set to 10 ± 3 mm/min, and on the load-deformation curve obtained, the starting point of deformation was from the point at which a preload of 222 N was applied [16].

The uniaxial compression creep test equipment for the shipping containers was composed of a hardware system consisting of a load-adding device, linear displacement-measuring device (linear variable differential transformer, LVDT), specimen compression device, load-measuring device, and software system to continuously measure and analyze the deformation of the specimen with time (Figure 2). In order to automatically measure the creep amount of the container continuously for a long-time by the LVDT installed in one place, the moving plate equipped with dead weight must perform accurate linear translation movement along the guide axis. For this purpose, a guide poster equipped with a ball bearing guide with a width of 10 cm was used in the creep test equipment manufactured. After applying a 222 N preload to set the reference point for measuring the deformation [16], three levels of constant load, selected from the ASTM D642 CS for each relative humidity scenario (23 °C; rh 50%, 70%, and 90%), were applied to the test specimen using a load-adding device; then, the 12 h creep test was performed [9].

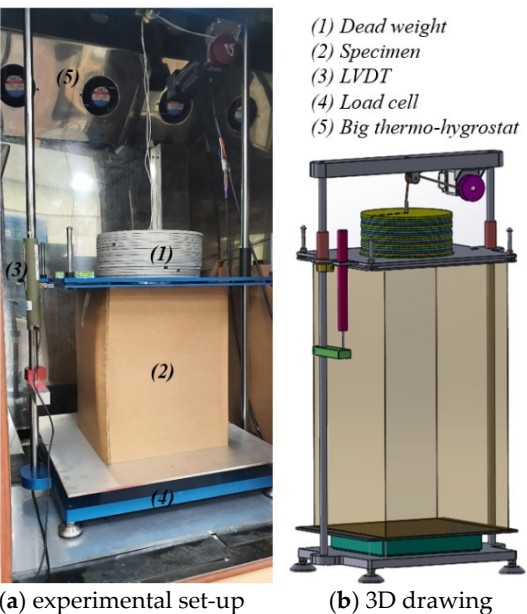

(1) Dead weight
(2) Specimen
(3) LVDT
(4) Load cell
(5) Big thermo-hygrostat

(**a**) experimental set-up        (**b**) 3D drawing

**Figure 2.** Uniaxial compression creep test apparatus for the shipping containers.

Before the compression and compressive creep tests, the samples were equilibrated under the planned temperature and humidity conditions for more than 48 h. Creep tests were conducted in a large chamber (L × W × H = 2.8 × 1.8 × 2.1 m), where the temperature and humidity were well maintained, with the freezer separated to reduce vibrations.

## 3. Results and Discussion

### 3.1. Uniaxial Compression Behavior Modeling for the Target Shipping Container

Figure 3 shows the stress-train (SS) curves obtained through the compression test for the target shipping container. Among grapes of the same color, a grape of a thick solid line represents the average for three repetitions. The maximum stress on the SS curve is CS, and the corresponding deformation is the failure strain (FS). In the case of a container, most of the compressive load is supported by the four vertical edges and side panels; therefore, the CS of the container is expressed as the load per unit peripheral length, considering the structural characteristics of the container [16].

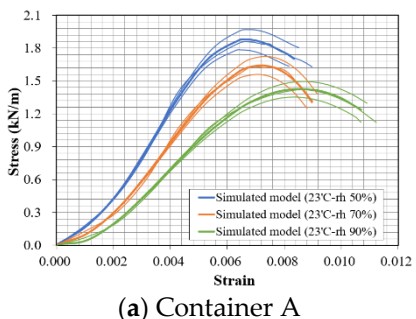       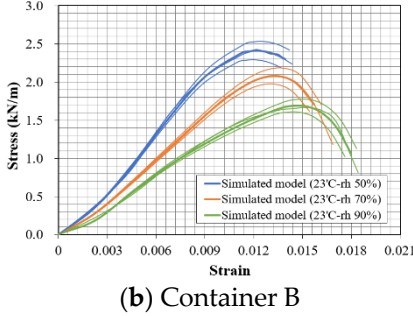       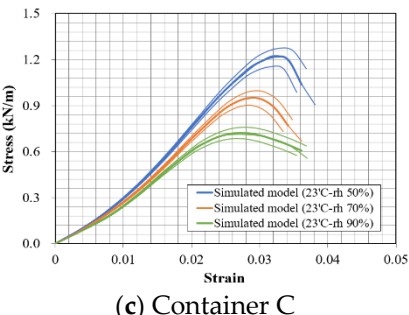

(**a**) Container A                    (**b**) Container B                    (**c**) Container C

**Figure 3.** Stress–strain curves for the target shipping containers.

Significant differences were observed in the CS and FS of the target shipping containers with respect to the relative humidity (Figure 3 and Table 4). In the case of the 'vertical long' Containers A and B, the CS in the standard condition decreased by approximately 15% and 24% as the relative humidity increased to 70% and 90%, respectively. In the case of the 'horizontal long' Container C, it decreased by approximately 24% and 42%, respectively (Table 4). As shown in Table 1, although the outer liners of the target shipping containers that come in direct contact with the humid outside air were the same (paperboard), the

primary cause of this difference is believed to be the difference in the geometric shape of the containers. In addition, in the case of Container C (Figure 3c), the FS decreased with the increase in the relative humidity; therefore, we deduced that the 'horizontal long' container is more sensitive to changes in the relative humidity.

**Table 4.** Compressive properties of the target shipping container according to the equilibrium condition [14].

| Division | Compression Strength (kN/m) | | | Failure Strain (mm/mm) | | |
|---|---|---|---|---|---|---|
| | rh 50% | rh 70% | rh 90% | rh 50% | rh 70% | rh 90% |
| Container A | 1.88 (0.14) [a] | 1.63 (0.17) [b] | 1.42 (0.19) [c] | $6.84 \times 10^{-3}$ $(3.53 \times 10^{-4})$ [a] | $7.70 \times 10^{-3}$ $(6.95 \times 10^{-4})$ [b] | $8.56 \times 10^{-3}$ $(9.41 \times 10^{-4})$ [c] |
| Container B | 2.44 (0.16) [a] | 2.07 (0.18) [b] | 1.85 (0.16) [c] | $1.26 \times 10^{-2}$ $(9.21 \times 10^{-4})$ [a] | $1.35 \times 10^{-2}$ $(1.28 \times 10^{-3})$ [b] | $1.53 \times 10^{-2}$ $(1.66 \times 10^{-3})$ [c] |
| Container C | 1.23 (0.13) [a] | 0.94 (0.08) [b] | 0.71 (0.08) [c] | $3.21 \times 10^{-2}$ $(1.85 \times 10^{-3})$ [a] | $3.01 \times 10^{-2}$ $(2.69 \times 10^{-3})$ [b] | $2.61 \times 10^{-2}$ $(2.85 \times 10^{-3})$ [c] |

Note: Mean comparison by Duncan's multiple range tests. [a, b, c] indicate the statistical difference in rows (significance level 5%).

Modeling each SS curve, as shown in Figure 3, was required to determine the CAE-predicted compression behavior in the multi-stacked packages, as applied to the target shipping containers. A procedure such as 3D response surface analysis is required to derive a model (3D, stress–strain-humidity) representing the more general SS behavior of a container from the experimental data (2D, stress–strain) in Figure 3. Figure 4 shows the results of the 3D response surface analysis of the SS behavior with respect to the relative humidity of each target shipping container. The modeling results are presented in Table 5. The shape of the response surface of stress–strain–relative humidity and the constant value of the model reflect the geometric shape of the container.

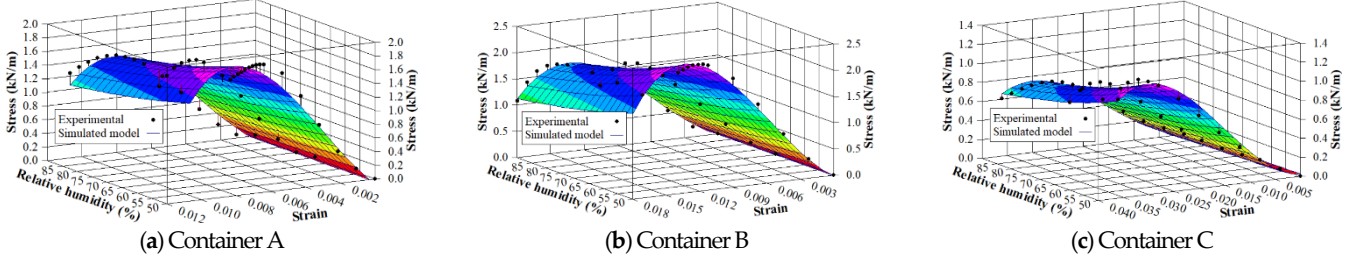

(**a**) Container A    (**b**) Container B    (**c**) Container C

**Figure 4.** Response surface analysis results for the SS behavior of the target shipping containers.

**Table 5.** SS behavior modeling results for the target shipping containers.

| Division | $\sigma = a(rh)^b (c + d\varepsilon + e\varepsilon^2 + f\varepsilon^3)$ $\sigma$ = Stress (kN/m), *rh* = Relative Humidity (%), $\varepsilon$ = Strain, *a–f* = Model Coefficients | | | | | | $r^2$ |
|---|---|---|---|---|---|---|---|
| | *a* | *b* | *c* | *d* | *e* | *f* | |
| Container A | 7.1770 | −0.5046 | $-8.1647 \times 10^{-2}$ | 220.7711 | 35,062.7922 | −3,964,056.2237 | 0.9703 |
| Container B | 16.9642 | −0.6407 | $-8.7621 \times 10^{-3}$ | 85.1669 | 17,046.9011 | −1,006,318.4112 | 0.9910 |
| Container C | 19.8104 | −0.7513 | $2.2976 \times 10^{-2}$ | 1.3531 | 3461.1754 | −76,603.4909 | 0.9819 |

### 3.2. Creep Behavior Modeling and Lifetime Prediction for the Target Shipping Container

Figure 5 shows the 12 h creep behavior with respect to the relative humidity and applied load of the target shipping container and the 3D response surface analysis results. Table 6 shows the modeling results for the 12 h creep behavior.

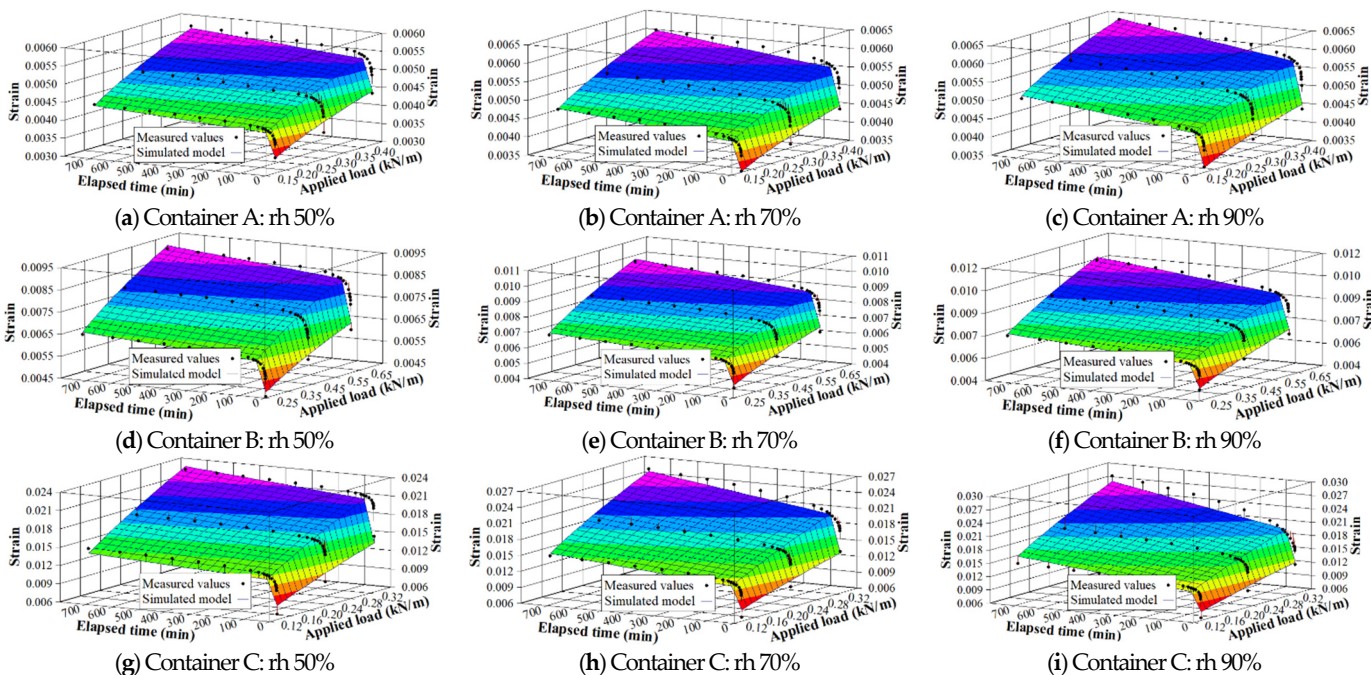

**Figure 5.** Response surface analysis results for 12 h creep behavior of the target shipping containers.

**Table 6.** The 12 h creep behavior modeling results for the target shipping containers.

| Division | rh (%) | $\varepsilon(L_o, t) = aL_o{}^b\{c + dt - exp(-et)\}$ <br> $\varepsilon$ = Strain, $L_o$ = Initial Static Load (kN/m), $t$ = Elapsed Time (min), $a-e$ = Model Coefficients | | | | | $r^2$ |
| | | *a* | *b* | *c* | *d* | *e* | |
| --- | --- | --- | --- | --- | --- | --- | --- |
| Container A | 50 | $0.1115 \times 10^{-2}$ | 0.3059 | 6.2218 | $0.6620 \times 10^{-3}$ | 0.2723 | 0.9394 |
| | 70 | $0.1261 \times 10^{-2}$ | 0.2817 | 5.5656 | $0.8130 \times 10^{-3}$ | 0.3213 | 0.9616 |
| | 90 | $0.1402 \times 10^{-2}$ | 0.2603 | 5.0795 | $0.9222 \times 10^{-3}$ | 0.3598 | 0.9761 |
| Container B | 50 | $0.2331 \times 10^{-2}$ | 0.3951 | 4.5110 | $0.5667 \times 10^{-3}$ | 0.2937 | 0.9847 |
| | 70 | $0.2637 \times 10^{-2}$ | 0.4223 | 4.2641 | $0.7352 \times 10^{-3}$ | 0.3641 | 0.9848 |
| | 90 | $0.2959 \times 10^{-2}$ | 0.4468 | 4.0470 | $0.8710 \times 10^{-3}$ | 0.4401 | 0.9852 |
| Container C | 50 | $1.3610 \times 10^{-2}$ | 0.5951 | 3.0018 | $0.5396 \times 10^{-3}$ | 0.7177 | 0.9552 |
| | 70 | $1.3073 \times 10^{-2}$ | 0.5743 | 2.9639 | $1.0357 \times 10^{-3}$ | 0.4683 | 0.9610 |
| | 90 | $1.2832 \times 10^{-2}$ | 0.5625 | 2.9003 | $1.5215 \times 10^{-3}$ | 0.3258 | 0.9336 |

Using the nonlinear creep model, as presented in Table 6, the 12 h creep behavior of the target shipping container is expressed in stages: instantaneous elastic strain that occurs within a very short time after the load is applied; retarded elastic creep, where the creep rate decreases with time; and steady-state creep, with an almost constant creep rate. The instantaneous elastic strain, retarded elastic creep, and steady-state creep rates increased as the level of applied load and relative humidity increased, and when the relative humidity at the same applied load was increased.

Based on the 12 h creep strain data, as shown in Figure 5, the 12 h average creep rate (CR$_{12}$) was analyzed using linear regression analysis, and the values were re-corrected up to failure using the method suggested by Burgers et al. [8,9]. Table 7 and Figure 6 show the corrected average creep rates (CR$_{FD}$). As an example, Figure 7 shows the analysis process of the CR$_{FD}$ until failure when an applied load of 0.1652 kN/m is used, at a relative humidity of 70%, for Container B. Thus, the 12 h creep strain was inferred from the nonlinear creep model shown in Table 6, while the analysis process of the CR$_{12}$ and CR$_{FD}$ up to failure is visually demonstrated.

**Table 7.** Lifetime prediction results of the target shipping containers [10].

| Division | rh (%) | Applied Load Level ($P$) (% $CS$) | $CR_{12}$ [(1)] (1/min) | Correction Factor, $P/(100-P)$ | $CR_{FD}$ [(2)] (1/min) | Lifetime (min) |
|---|---|---|---|---|---|---|
| Container A | 50% | 9.3084 | $7.8404 \times 10^{-7}$ | 0.1027 | $8.0473 \times 10^{-8}$ | 85,000 |
| | | 15.9574 | $9.4011 \times 10^{-7}$ | 0.1899 | $1.7850 \times 10^{-7}$ | 38,300 |
| | | 22.6064 | $1.0983 \times 10^{-6}$ | 0.2922 | $3.2081 \times 10^{-7}$ | 21,300 |
| | 70% | 10.7362 | $9.9283 \times 10^{-7}$ | 0.1203 | $1.1941 \times 10^{-7}$ | 64,500 |
| | | 18.4049 | $1.1839 \times 10^{-6}$ | 0.2255 | $2.6705 \times 10^{-7}$ | 28,800 |
| | | 26.0736 | $1.3832 \times 10^{-6}$ | 0.3526 | $4.8785 \times 10^{-7}$ | 15,800 |
| | 90% | 12.3239 | $1.1962 \times 10^{-6}$ | 0.1405 | $1.6814 \times 10^{-7}$ | 50,900 |
| | | 21.1368 | $1.4276 \times 10^{-6}$ | 0.2679 | $3.8239 \times 10^{-7}$ | 22,400 |
| | | 29.9296 | $1.6681 \times 10^{-6}$ | 0.4271 | $7.1251 \times 10^{-7}$ | 12,000 |
| Container B | 50% | 10.3934 | $1.5357 \times 10^{-6}$ | 0.1159 | $1.7813 \times 10^{-7}$ | 70,700 |
| | | 17.8156 | $1.8428 \times 10^{-6}$ | 0.2168 | $3.9947 \times 10^{-7}$ | 31,500 |
| | | 25.2377 | $2.1827 \times 10^{-6}$ | 0.3376 | $7.3682 \times 10^{-7}$ | 17,100 |
| | 70% | 12.2512 | $1.8010 \times 10^{-6}$ | 0.1396 | $2.5145 \times 10^{-7}$ | 53,600 |
| | | 21.0000 | $2.2129 \times 10^{-6}$ | 0.2658 | $5.8824 \times 10^{-7}$ | 22,900 |
| | | 29.7488 | $2.7047 \times 10^{-6}$ | 0.4235 | $1.1453 \times 10^{-6}$ | 11,800 |
| | 90% | 13.7081 | $2.0664 \times 10^{-6}$ | 0.1589 | $3.2826 \times 10^{-7}$ | 46,500 |
| | | 23.4973 | $2.5830 \times 10^{-6}$ | 0.3072 | $7.9335 \times 10^{-7}$ | 19,300 |
| | | 33.2865 | $3.2288 \times 10^{-6}$ | 0.4990 | $1.6110 \times 10^{-6}$ | 9480 |
| Container C | 50% | 10.9593 | $5.4254 \times 10^{-6}$ | 0.1231 | $6.6777 \times 10^{-7}$ | 48,100 |
| | | 18.7967 | $5.0598 \times 10^{-6}$ | 0.2315 | $1.1712 \times 10^{-6}$ | 27,400 |
| | | 26.6260 | $4.8542 \times 10^{-6}$ | 0.3630 | $1.7615 \times 10^{-6}$ | 18,200 |
| | 70% | 14.3404 | $5.4809 \times 10^{-6}$ | 0.1674 | $9.1757 \times 10^{-7}$ | 32,800 |
| | | 24.5957 | $6.5818 \times 10^{-6}$ | 0.3263 | $2.1469 \times 10^{-6}$ | 14,000 |
| | | 34.8404 | $1.1741 \times 10^{-5}$ | 0.5347 | $6.2778 \times 10^{-6}$ | 4800 |
| | 90% | 18.9859 | $5.5365 \times 10^{-6}$ | 0.2344 | $1.2975 \times 10^{-6}$ | 20,100 |
| | | 32.5634 | $8.1000 \times 10^{-6}$ | 0.4828 | $3.9113 \times 10^{-6}$ | 6670 |
| | | 46.1268 | $1.8628 \times 10^{-5}$ | 0.8563 | $1.5949 \times 10^{-5}$ | 1640 |

Note: (1) Average creep rate in 12 h test. (2) Average creep rate over the entire time up to failure at load level *P*.

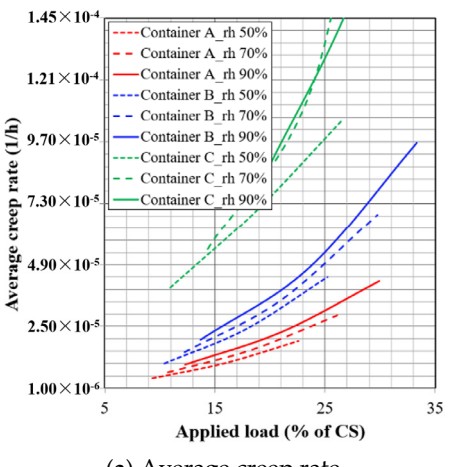

(**a**) Average creep rate

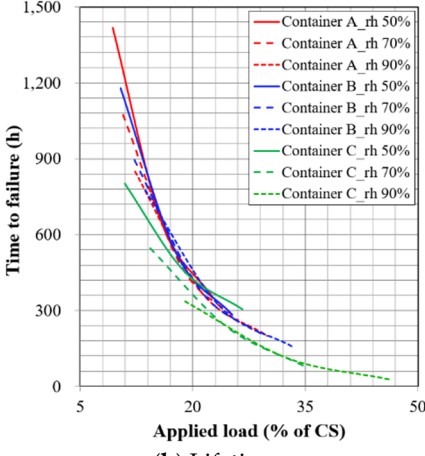

(**b**) Lifetime

**Figure 6.** Average creep rate and lifetime with respect to relative humidity and applied load for the target shipping containers.

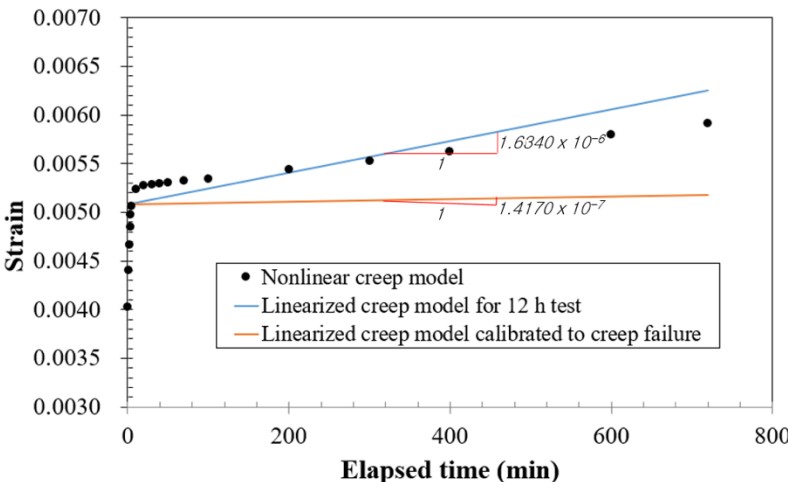

**Figure 7.** Example of the average creep rate from a nonlinear creep model: Container B, rh = 70%, $L_o$ = 0.1652 kN/m, applied load level (% of CS) = 0.1652/2.07 = 7.98%, correction factor = 7.98/(100−7.98) = 0.0867 (Tables 4 and 6).

As the applied load increased at all relative humidity levels in this study, the rate of increase in the $CR_{FD}$ was higher Except for 'horizontal long' container C, which has a unique structure compared to containers A and B, the higher the applied load level, the greater the difference in average creep rate by relative humidity. In general, it was found that the average creep rate of the container was more affected by the applied load than the relative humidity (Figure 6a). The lifetime (time to failure) estimated from the FS and average creep rate of the container gradually slowed as the applied load increased at all relative humidity levels (Table 7, Figure 6b). However, as the applied load level increased, the difference with respect to the relative humidity tended to decrease. Overall, the effect of the applied load level was greater than that of relative humidity. For example, in the case of Container B, when applied loads of approximately 10.4%, 17.8%, and 25.2% of the CS in the standard condition (relative humidity 50%) were applied, the predicted lifetimes decreased by approximately 24%, 27%, and 31%, respectively, compared to the lifetime of the standard condition at 70% relative humidity, and by approximately 34%, 39%, and 45%, respectively, at 90% relative humidity.

*3.3. Load-Sharing Rate of Target Package*

Accurate calculation of the load-sharing rate between the shipping container and cushioning system in packages equipped with the EPS-based cushioning system is important to achieve the proper packaging design of the product. As the stacking load, excluding the load-sharing rate of the shipping container, is transmitted to the product through the cushioning material, the strength independence of the product is used as the basis for designing the shipping container. In other words, if the strength independence of the product is zero, the shipping container shall be responsible for all stacking loads. However, if there is a certain level of strength independence owing to the nature of the product, the stacking load, excluding this strength independence, becomes the basis of the strength design of the shipping container.

Figure 8 shows the analysis results of the load-sharing rate acting on the shipping container and the EPS cushioning material of the bottom layer package for each stacking situation (Table 1) of the target packages. The SS models for the target shipping containers in Table 5 and the SS models of the EPS previously published by the researchers [17] were applied to this estimation.

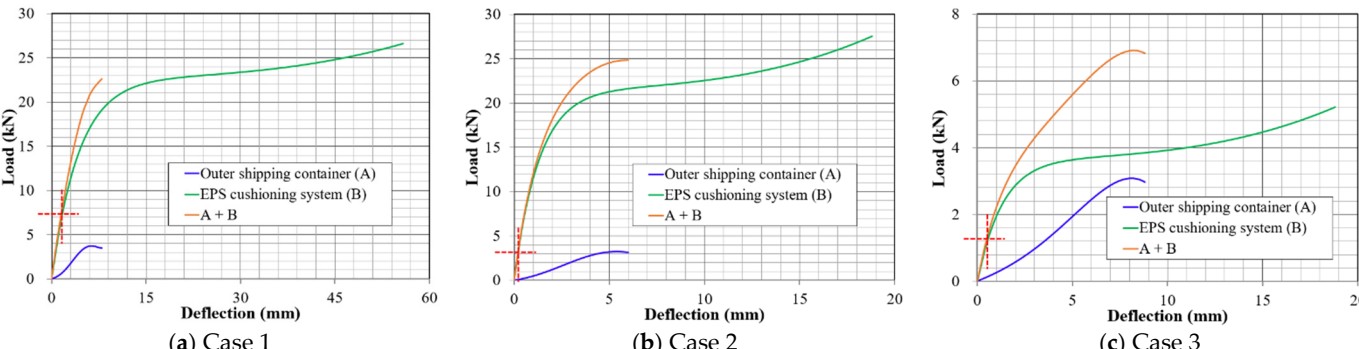

**Figure 8.** Calculation of load-sharing rate from SS curves of the shipping container and EPS cushioning material.

In the target packages used in this study, the ratio of the load-sharing rate between the shipping container and EPS cushioning material was determined to be about 2%:98%, with most of the stacking load applied to the product through the cushioning material. Therefore, because the compression behavior of the target package depends on the compression behavior of the cushioning material in the package as opposed to the shipping container, data on the compression and compressive creep behavior of the EPS-based cushioning system are considered to be more important in CAE prediction technology for determining the compression behavior of multi-stacked target packages.

## 4. Conclusions

CAE prediction technology for compression behavior in multi-stacked packages is an important tool for preventing warehouse accidents, shortening packaging development periods, and optimizing packaging. In this study, compression and compressive creep behavior of target shipping containers, which are material properties based on FEA, and the lifetime and load-sharing rate, were analyzed to develop CAE prediction technology for predicting the multistage compression behavior of three target packages with different logistics conditions. The research results can be summarized as follows:

1. The reduction rate of the compression strength of the container with respect to the increase in the relative humidity was greater in the 'horizontal long' container than in the 'vertical long' container. However, the failure strain increased with increasing relative humidity in the 'vertical long' container, but decreased in the 'horizontal long' container. A nonlinear regression model for the SS behavior of the target shipping container was developed for the relative humidity and strain.

2. Using a 12 h creep experiment on the target shipping containers, a nonlinear creep model was developed with the applied stress and elapsed time as factors of the model. The instantaneous elastic strain, retarded elastic creep, and steady-state creep rates increased as the load was applied and the relative humidity increased, and when the relative humidity at the same applied load increased.

3. Based on the data obtained from the 12 h creep strain experiment, the 12 h average creep rate ($CR_{12}$) was analyzed using linear regression analysis, and the values were re-corrected until failure. As the applied load increased, the rate of increase in the average creep rate ($CR_{FD}$) increased, i.e., the higher the applied load level, the larger the difference in the average creep rate with respect to the relative humidity.

4. The lifetime (time to failure) estimated from the failure strain and average creep rate of the container gradually decreased as the applied load increased at all values of relative humidity. However, as the applied load increased, the difference with respect to the relative humidity tended to decrease.

5. In the target packages used in this study, the ratio of the load-sharing rate between the shipping container and EPS cushioning material was determined to be 2%:98%, with most of the stacking load applied to the product through the cushioning material. Therefore, data on the compression and compressive creep behavior of the EPS-based cushioning system are considered significant in developing a CAE prediction technology for the compression behavior of multi-stacked target packages.

**Author Contributions:** Data curation, J.M.P. and H.M.J.; formal analysis, J.M.P; funding acquisition, H.M.J.; investigation, G.Y.L. and D.H.K.; software, J.M.P. and H.M.J.; validation, J.M.P.; visualization, J.M.P.; writing—original draft, J.M.P. and H.M.J.; writing—review and editing, J.M.P., G.Y.L. and H.M.J.; supervision, H.M.J. All authors have read and agreed to the published version of the manuscript.

**Funding:** This research was carried out with the support of Cooperative Research Program for development of data application technology for postharvest management of the agricultural and livestock product (Project No.: PJ017050042022), Rural Development Administration.

**Institutional Review Board Statement:** Not applicable.

**Informed Consent Statement:** Not applicable.

**Data Availability Statement:** MDPI Research Data Policies.

**Conflicts of Interest:** The authors declare no conflict of interest.

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
