# Peer review of "CAE Prediction for Compression Behavior in Multi-Stacked Packages with an EPS-Based Cushioning System: Modeling of Compression and Compressive Creep Behavior"

_applsci, doi:10.3390/app13042215_

Round 1
Reviewer 1 Report
1. Recommendation:
Minor revision (re-review is not required).
2. Comments to Author:
Title: CAE prediction for compression behavior in multi-stacked packages with EPS-based cushioning system: modeling of compression and compressive creep behavior.
This is a very important and meaningful article. The main goal of this paper is to setup experiment model and finite element model for analysis of compression and compressive creep of target shipping containers. The numerical results are useful for design, testing and produce the shipping containers. The manuscript is well written and the content is merit, therefore, the paper is acceptable after some minor corrections:
1. The font in the paper is not unified. Please check whether the format of the paper is correct (Please check all tables and lines: 159-169, 277-285; and several other positions).
2. Punctuation was unified: decimal points and decimal commas (Table 7).
3. Following errors should be corrected:
- Figure 5: “(g) Container C: rh 90%” à “(g) Container C: rh 50%”
- Line 205: “(Figure 3(c)]” à “[Figure 3(c)]”
4. The physical meaning of Figure 4, 8 should be more discussed throughout.
Reviewer 2 Report
Article has relevance in the area presented, and can be further explored in the field of mathematics by applying finite elements....Explore more recent research to improve the composition of the research.... this approach closes with the possible gains from the study.
Improve the conclusion by showing the experimental results indicating the possible gains.
Reviewer 3 Report
In this manuscript, the authors studied the compressive creep behavior of shipping containers with finite element modeling approach. The authors stated that they utilized finite element analysis to simulate the mechanical response of the testing materials and derived the relationship between stress, strain, and relative humidity. The lifetime of the testing materials is predicted and reported for various containers and testing conditions. Before the paper can be accepted, there are some questions needed to be answered and addressed.
1, in the first paragraph of the introduction, if a schematic figure can be added to demonstrate the cushioning system in the package, it might be much easier for readers to understand the structures.
2, Line 57, the formula here is recommended to be listed as a separate line.
3, Line 108, please differentiate the applied load level P and the load P.
4, Line 112, references should be added here about the related studies using experiments or CAE about the compression behavior of similar systems.
5, Table 1, "total weigth" -> "total weight".
6, Table 1, which part does cushion area and cushion thickness refer to in the schematic figure? It is hard to tell. The authors are recommended to either using arrows and texts to mark in the schematic, or add some explanation in the context.
7, Table 1, why does the schematics has header "cushioning method"?
8, Line 143, what does prototype mean here? Is it the real system that the modeling is based upon? Some clarifications are needed.
9, Table 2, the abbreviation "RSC", "BB/F-DW", "A/F-SW" may need some explanations.
10, Line 180, for abbreviations such as "rh", "LVDT", the authors are recommended to add full words at their first usage.
11, It is confusing for section 2.2, is this manuscript about experiments, or modeling (or both)? If it is about experiments, then experimental results should be reported as well; if it is about modeling, then the necessary information about modeling approaches and modeling methods should be described in detail, especially for FEA, including all the input parameters, materials' physical parameters, creep model used, modeling software (or code), meshing, PDE solvers used, etc.
12, Figure 3, why are there multiple curves for each modeling condition? Are they coming from different modeling jobs? If so, why do the results vary for the same condition and the same container? What is the reason for different results?
13, Line 221, analysis based on Figure 4 should be added here, such as the observations from this figure.
14, Figure 4, color bar is missing.
15, Are the simulated results in Figure 4 coming from FEA modeling, or from data-fitting using formula in Table 5?
16, Where does the formula in Table 5 come from? Any reference for it? Are the coefficients coming from data fitting using experimental data? Are there any physical meanings for the coefficients? How to interpret them and compare these coefficients among different containers?
17, Figure 5, color bar is missing.
18, Table 6, similar questions as # 16.
19, Line 261, any paragraphs are about Figure 6(b)?
20, Figure 6(a), why are the data of Container C not plotted in Figure 6(a)?
21, Are there references with similar studies that the authors can compare the results with? If so, please add them.
Reviewer 4 Report
The present manuscript was generate a model of compression and compressive creep behavior of multi stack packages through FE software. The introduction part was written well, however the number of references and literature review was not enough. The methodology and experimental procedure is not explain well. One of the most effective factors in this study was humididity induced process and the authors missed to explain how they applied rh to they test setup. Also, the position of LVDT is not appropritate for this test and it is not based on ASTM D642 CS procedure. At least two LVDTs should be used in this kind of tests and the average of displacement should be rcorded. This is much vital when a material such as EPS with localized buckling are used. In figure 6 the valuable information can be seen. It is expected that more explanation about load dependency and rh dependency of material is presentd. Moreover from figure 6a more detail about creep strain rate should be expressed. Figure 8 needs a higher quality to published.
I look forward to receive the revised manuscript after minor corrections.
Round 2
Reviewer 3 Report
After the revision, the quality of the manuscript gets improved greatly. All questions and concerns were properly addressed in the revised version. In response 2, the authors stated that the expression of Line 57 is listed on a separate line, but may forget to.
There seems to be no similar studies on this topic, which makes this paper appealing to readers. I recommend the manuscript to be accepted.
Reviewer 4 Report
The responses are satisfactory and the paper can be published in the present form.